# Antimicrobial, Antigenotoxicity, and Characterization of *Calotropis procera* and Its Rhizosphere-Inhabiting Actinobacteria: In Vitro and In Vivo Studies

**DOI:** 10.3390/molecules27103123

**Published:** 2022-05-13

**Authors:** Amna A. Saddiq, Hend M. Tag, Nada M. Doleib, Asmaa S. Salman, Nashwa Hagagy

**Affiliations:** 1Department of Biology, College of Science, University of Jeddah, Jeddah 21959, Saudi Arabia; 2Department of Biology, College of Science and Arts at Khulis, University of Jeddah, Jeddah 21959, Saudi Arabia; hmtaha@uj.edu.sa (H.M.T.); ndoleib@uj.edu.sa (N.M.D.); assalman@uj.edu.sa (A.S.S.); 3Zoology Department, Faculty of Science, Suez Canal University, Ismailia 41522, Egypt; 4Department of Microbiology, Biotechnology and Biochemistry, Faculty of Applied and Industrial Sciences, University of Bahri, Kartoum 13311, Sudan; 5Genetics & Cytology Department, Biotechnology Research Institute, National Research Center, Giza 12622, Egypt; 6Botany and Microbiology Department, Faculty of Science, Suez Canal University, Ismailia 41522, Egypt

**Keywords:** *Calotropis procera*, actinobacteria, antimicrobial activity, genotoxicity, DNA fragmentation, chromosomal aberration

## Abstract

*Calotropis procera* (*C. procera*) is a wild shrub that is a medicinal plant found in abundance throughout Saudi Arabia. In this study, we investigated the phytochemical composition and antigenotoxic properties of the ethanolic extract of *C. procera*, in addition to the antimicrobial activity of the plant and its rhizospheric actinobacteria effects against pathogenic microorganisms. Soil-extract medium supplemented with glycerol as a carbon source and starch–casein agar medium was used for isolation of actinobacteria from rhizosphere. From the plant, a total of 31 compounds were identified using gas chromatography/mass spectrometry (GC–MS). The main components were α-amyrin (39.36%), lupeol acetate (17.94%), phytol (13.32%), hexadecanoic acid (5.55%), stigmasterol (3.16%), linolenic acid (3.04%), and gombasterol A (2.14%). *C. procera* plant extract’s antimicrobial activity was investigated using an agar well-diffusion assay and minimum inhibitory concentration (MIC) against six pathogenic microbial strains. The plant extract of *C. procera* was considered significantly active against *Staphylococcus aureus*, *Klebsiella pneumonia*, and *Escherichia coli*, with inhibition zones of 18.66 mm, 21.26 mm, and 21.93 mm, respectively. The plant extract was considered to be a moderate inhibitor against *Bacillus subtilis*, with MIC ranging from 0.60–1.50 mg/mL. On the other hand, the isolated actinobacteria were considered to be a moderate inhibitor against *S. aureus* (MIC of 86 µg/mL), and a potent inhibitor, strain CALT_2, against *Candida albicans* (MIC of 35 µg/mL). The 16S rRNA gene sequence analysis showed that the potential strains belonged to the genus *Streptomyces.* The effect of *C. procera* extract against cyclophosphamide (CP)-induced genotoxicity was examined by evaluating chromosome abnormalities in mouse somatic cells and DNA fragmentation assays. The current study revealed that oral pretreatment of *C. procera* (50, 100, and 200 mg/kg b.w.) for 1, 7, and 14 days to cyclophosphamide-treated animals significantly reduced chromosomal abnormalities as well as DNA fragmentation in a dose-dependent manner. Moreover, *C. procera* extract had antimicrobial and antigenotoxic effects against CP-induced genotoxicity.

## 1. Introduction

*Calotropis procera* (Aiton) Dryand is a perennial, soft-wooded shrub, belonging to the Apocynaceae family and subfamily *Asclepiadaceae*. This evergreen, xerophytic plant thrives in dry and semiarid environments. In different regions of the world, it is known by numerous common names such as apple of Sodom, calotrope, wild cotton, Indian milkweed, gigantic milkweed, and rubber tree, and in Saudi Arabia, it is called “Ushar”. In North Africa, the Middle East, South Asia, and Southeast Asia, it has long been utilized in traditional medicinal applications. Since antiquity, it has been used for fuel, fiber, feed, and lumber [1,2,3].

Within salt-stressed environments, endophytic bacteria such as *Virgibacillus koreensis* and *Pseudomonas stutzeri* have been found to be associated with *C. procera*, which may help it survive under harsh conditions. In addition, endophytic fungal species, such as *Phaeoramularia calotropidis*, *Curvularia hawaiiensis*, *Guignardia bidwellii*, *Alternaria alternata*, *Cochliobolus hawaiiensis*, *Aspergillus* spp., *Mucor circinelloides*, *Fusarium* spp., *Chaetomium* spp., and *Penicillium* spp., provide protection for the plant against pathogens and pests [4,5].

Flavonoids, terpenoids, alkaloids, tannins, saponins, cardiac glycosides, and steroids have been found in several portions of the plant, according to several studies [6,7,8,9]. Fatty acid ethyl esters (21.4%), palmitic acid esters (10.2%), amino acids (8.1%), and linoleic acids (7.4%) are the principal phytochemical groups found in *C. procera* leaf extracts [10].

In several plant and animal cells, including human cells, *C. procera* causes acute toxicity. As a result, numerous plant parts, particularly latex, have been tested against various cancer cell lines [11,12,13]. Similarly, the plant’s antibacterial and anthelmintic properties are being explored in pharmacology. The toxicity–bioactivity relationship of *C. procera*, on the other hand, has yet to be well examined. According to a few studies, the plant causes acute cardiotoxicity and hepatotoxicity [14]. These toxic effects have not been investigated in detail, and additional research is needed to confirm the therapeutic potential of *C. procera*. The search for environmentally friendly prototypes to replace chemically manufactured pharmaceuticals is on the rise. As a result, several studies have been conducted on the plant species described in traditional medical systems. The pharmacological capabilities of *C. procera* have been used to treat a variety of various human infections in the past, including colds, fevers, leprosy, rheumatism, asthma, indigestion, eczema, elephantiasis, diarrhea, dysentery, and skin diseases [8]. In Saudi Arabia, a decoction of aboveground portions is used to cure fever, joint pain, constipation, and muscle spasms [6]. In Burkina Faso, the plant is also used to treat mental conditions [15]. Secondary metabolites and cardiotonic compounds found in *C. procera* are responsible for the therapeutic properties of the plant [16,17].

In previous studies, Mossa et al. and Garabadu et al. [6,18] found that the extracts of the aboveground plant sections of *C. procera* have high antipyretic, antidepressive, analgesic, and neuromuscular-blocking activities. Antibacterial activity was observed in extracts from the bark and leaves against *Pseudomonas aeruginosa*, *Klebsiella pneumoniae*, *Escherichia coli*, and *Bacillus subtilis* [17]. The extracts of both aerial portions of *C. procera* and its endophytic bacteria, *Bacillus siamensis*, have been demonstrated to have a broad antibacterial range [16]. *C. procera* leaf extracts also significantly lower blood glucose levels, demonstrating their antihyperglycemic potential [19]. Although the pharmaceutical and industrial applications of the plant have attracted considerable interest, the plant’s biological and ecological characteristics (especially those focusing on adaptations or plasticity) have received little study in general. Furthermore, the toxicity–bioactivity relationship of *C. procera* has not been well studied, which is important for verifying its therapeutic properties. Evaluating these fundamental aspects could help *C. procera* become more commercially viable and open it up to new applications. Moreover, filling in these knowledge gaps could benefit a better understanding of its invasive behavior and potential future biodiversity and/or environmental problems [20].

In this study, the antimicrobial activity of *C. procera* extract was evaluated, along with the antimicrobial activity of its rhizosphere-inhabiting actinobacteria, which was evaluated for the first time, to our knowledge. Moreover, the present investigation aimed to evaluate the cytogenetic bioactivity of *C. procera* leaves to ameliorate the cytogenetic alterations and DNA damage induced by cyclophosphamide (CP), an alkylating agent used as a potent anti-inflammatory and immunosuppressive cytostatic and cytotoxic drug to treat diverse medical problems such as neoplasia.

## 2. Results

### 2.1. Characterization and Identification of the Potential Actinobacterial Isolates

Based on colony shape, color, and texture, a total of 17 isolates were selected for preliminary screening of antimicrobial activity. Only four isolates, designated as CALT_1, CALT_2, CALT_3, and CALT_4, showed potential activity and were selected for further investigation. A 16S rRNA gene-sequence analysis showed that the potential actinobacterial strains were affiliated within genus *Streptomyces*, with similarity ≥ 97%, and closely related to *Streptomyces coeruleorubidus*, *Streptomyces maritimus*, *Streptomyces carminius*, and unclassified species within the same genus, as shown in Figure 1. The 16S rRNA gene data were deposited under the accession numbers MT742093-MT742096 in the NCBI and GenBank nucleotide sequence databases.

### 2.2. Antimicrobial Activity

In the present study, the extract from *C. procera* was examined for antimicrobial activity quantitatively at a concentration of 10 mg/mL (100 µg) by zone inhibition on an agar plate (Table 1). The results revealed that the ethanolic extract of *C. procera* possessed potential antibacterial activity. *C. procera* leaf extract showed significant activity against all tested microorganisms compared with the standard antibiotics gentamycin and ketoconazole. The most antibacterial activity was recorded in *S. aureus* and *K. pneumoniae*, followed by *B. subtilis* and *E. coli*. The antifungal activity of leaf extract of *C. procera* was also significant against the tested pathogenic fungi *C. albicans* and *A. fumigatus* compared with ketoconazole. Regarding the effectiveness of the tested plant extract against Gram-negative bacteria, the results revealed that the significant inhibition zone of 21.93 ± 1.71 mm was observed against *E. coli* which was comparatively insignificant compared to the positive control (23.40 ± 2.42 mm). In addition, *C. procera* showed potent activity against *K. pneumoniae* (ZOI = 21.26), which was not significantly different from that of the reference antibiotic. These values fall within the range considered to be highly sensitive when compared to the control antibiotic.

The minimum inhibitory concentration (MIC) results of *C. procera* extract against different pathogenic microorganisms are shown in Figure 2, Figure 3 and Figure 4. The ethanolic extract of *C. procera* leaves showed significantly high MIC values for both *C. albicans* and *A. fumigatus* compared with the actinobacterial extract from strains CALT_1, CALT_2, CALT_3, and CALT_4 (Figure 2). Figure 3 displays the MIC for the tested extract against Gram-positive bacteria; the current results showed that CALT_1, CALT_2, CALT_3, and CALT_4 were more effective than the *C. procera* leaf extract. Regarding the effectiveness of tested extracts against Gram-negative bacteria, *C. procera* extract revealed potent antimicrobial activity against *E. coli* as compared with CALT_1, CALT_2, CALT_3, and CALT_4, while CALT_2 exhibited more effectiveness against *Klebsiella pneumoniae*.

Regarding the antimicrobial activity of rhizosphere-inhabiting actinobacteria, out of 17 actinobacterial isolates screened for antimicrobial activity, 4 exhibited potential antimicrobial activities against tested pathogens, with inhibitory zone diameters ranging from 6.5 to 13.7 mm shown by strain CALT_4 against *B. subtilis* and strain CALT_2 against *Candida albicans*, respectively, as shown in Table 1. The extract of strain CALT_2 showed the most potent activity against tested pathogens, with MIC values of 35 µg/mL against *A. fumigatus* as shown in Figure 2, 56 µg/mL against *B. subtilis* as shown in Figure 3, and 54 µg/mL against *K. pneumoniae* as shown in Figure 4.

According to Figure 2, the MIC results revealed that the ethanolic extract of *C. procera* leaves showed significantly high MIC values against both *C. albicans* and *A. fumigatus* as compared with the actinobacteria extracts CALT_1, CALT_2, CALT_3, and CALT_4. Figure 3 displays the MICs for the tested extracts against Gram-positive bacteria, and the current results showed that CALT_1, CALT_2, CALT_3, and CALT_4 were more effective than the *C. procera* leaf extract. Regarding the effectiveness of the tested extracts against Gram-negative bacteria, *C. procera* extract revealed potent antimicrobial activity against *E. coli* as compared with strains CALT_1, CALT_2, CALT_3, and CALT_4, while CALT_2 exhibited more effectiveness against *Klebsiella pneumoniae*.

### 2.3. Gas Chromatography–Mass Spectrometry (GC–MS)

A phytochemical study was carried out by GC–MS analysis of the *C. procera* leaf extract. The chromatogram identified 31 phytochemicals as constituents (Figure 5), of which α-amyrin was the major compound (39.36%) identified at retention time 63.63 min, followed by lupeol acetate (17.94%) at retention time 64.65 min, phytol (13.32%) at retention time 36.71 min, hexadecanoic acid (5.55%) at retention time 32.41 min, stigmasterol (3.16%) at retention time 55.61 min, and linolenic acid (3.04%) at retention time 38.12 min. The remaining constituent chemical compounds were present in proportions of less than 2%. The components and their retention times, molecular formulas, and molecular weights are summarized in Table 2. Figure 6 shows the demonstrated hit spectrum and chemical structure of the major compounds present in *C. procera* leaf extract. The analysis of extract from *Streptomyces* sp. strain CALT_2 by GC–MS led to the detection of three compounds on the basis of retention time and mass analysis (Table 3). The following compounds were identified: (i) hexadecanoic acid, (ii) stigmasterol, (iii) α-amyrin. Spectra and chemical structure are presented in Figure 6.

### 2.4. In Vivo Studies of C. procera

#### Acute Toxicity Test

The toxicity of the ethanolic extract of *C. procera* was assessed at different doses of up to 5000 mg/kg. It was observed that the tested extract did not cause any mortality or changes in the behavior of the treated mice from the beginning of administration until 14 days. No changes were observed in food or water intake, which confirmed that the extract was safe and did not cause any toxicity, even at high doses.

### 2.5. Chromosomal Aberrations in Bone Marrow Cells

Table 4 shows the total counts and percentages of chromosomal aberrations in control and *C. procera*-treated animals. The percentage of chromosome aberrations in animals treated with single and repeated doses of *C. procera* was not significantly different from that in the control animals (Table 4). No significant reduction in chromosomal abnormalities induced by CP was observed after a single treatment with *C. procera*. Repeated treatment with *C. procera* for 7 and 14 days caused a significant (*p* < 0.01) reduction in the percentage of chromosomal abnormalities induced by CP (Table 5). The percentage of reduction reached 42.85% and 58.3% after pretreatment with *C. procera* for 7 and 14 days, respectively. Table 5 illustrates the protective effect of *C. procera* in reducing the different types of aberrations.

### 2.6. DNA Fragmentation

Administration of single and repeated doses of *C. procera* caused no significant DNA fragmentation (Table 6). Pretreatment with a repeated dose of *C. procera* significantly (*p* < 0.01) decreased the percentage of DNA fragmentation induced by CP in liver cells (Table 7). The percentage of DNA fragmentation was reduced to 5.34% and 4.29% (*p* < 0.01) after pretreatment with *C. procera* for 7 and 14 days, respectively, compared with 8.77% for the groups treated only with CP.

## 3. Discussion

The current study investigated the in vitro antimicrobial activity against some pathogenic microorganisms and the in vivo antigenotoxicity of *C. procera* leaf ethanolic extract. The evolving resistance of pathogenic microbes to currently existing antimicrobial agents requires new antimicrobial agents. The use of medicinal plants as a natural alternative is the primary research field for overcoming drug resistance to infectious agents. Scientists still need to assess medicinal plants’ effectiveness against microbes [21,22,23]. Several activities have been attributed to *C. procera*, including antibacterial [24], antifungal [25], and antitumoral [26], which indicate the pronounced biological potential of this genus. In the present study, the phytochemical constituents of the *C. procera* ethanolic extract from leaves were evaluated. Of these, 39.36% were α-amyrin esters, suggesting that they may be chemical markers for *C. procera*. Lupeol acetate, phytol, hexadecanoic acid, stigmasterol, and linolenic acid were also identified. The current finding of many medicinal plants for bioactive potential has generated a growing interest in the bioactive potential of their soil-inhabiting microbes. The antibacterial activity of *Streptomyces* sp. strain CALT_2, an actinobacterium found in the rhizosphere of *C. procera*, was investigated in this study. The GC–MS analysis of the bacterial extract identified three chemicals that were also found in the leaf extract, but their relative concentrations in the bacterial extract were higher, implying that microbial communities may contribute to the principal active component generated in plant tissue.

The results of this study revealed that the hydroethanolic extract of *C. procera* leaves showed potent antifungal activity against *Candida albicans* and *Aspergillus fumigatus* compared with the standard drug, ketoconazole, a broad-spectrum antifungal medication. Moreover, *C. procera* exhibited potent antibacterial activity against Gram-positive bacteria *S. aureus* and Gram-negative bacteria *K. pneumoniae* compared with gentamycin, an aminoglycoside antibiotic effective against a wide range of pathogenic bacteria. Previous studies on the antipathogenic activity of the methanolic extract of *C. procera* leaves have shown its potential against *S. aureus* and *S. typhi* [27]. This finding was in agreement with that of the current study. The plant extract’s ability to destroy or inhibit the growth of pathogenic microbes with excellent efficiency indicates the presence of bioactive secondary metabolites that have been considered to be antimicrobial agents [28]. Moreover, Thenmozhi et al. [29] stated that the antibacterial activity of plants is due to the secondary metabolites they form for protection against pests, herbivores, and microbial infections.

The distinct antimicrobial activity of *C. procera* could be attributed to the presence of α-amyrin, a pentacyclic triterpene. Triterpenes have important antimicrobial properties, as reported in previous studies [30,31]. Previous results supported our findings. Johann et al. [32] investigated the antifungal activity of amyrin against *Candida* species. Singh and Dubey [33] demonstrated that β-amyrin acetate isolated from *Heliotropium marifolum* showed potent activity against *Penicillium chrysogenum*, *E. coli*, and *K. pneumoniae*. According to Awolola et al. [34], lupeol acetate exhibited moderate antimicrobial activity against *S. aureus*. Saha et al. [35] demonstrated that phytol was a potent antimicrobial agent.

On the other hand, among the species with the most potential for producing biologically active compounds are those of *Streptomyces*, which also play a significant role in the protection of plants against pathogens [36]. However, the antimicrobial potential of actinobacteria inhabiting rhizosphere soil from this plant has not been investigated previously. In this investigation, four species belonging to genus *Streptomyces* isolated from the rhizosphere of a common medicinal plant in Saudi Arabia, *C. procera*, showed significant antimicrobial activity against two Gram-negative bacteria (*E. coli* and *Klebsiella pneumonia*), two Gram-positive bacteria (*S. aureus* and *B. subtilis*), one yeast (*Candida albicans*), and one fungus (*Aspergillus fumigatus*). The results indicated that *Streptomyces carminius* strain CALT_4 showed the least inhibitory activity (6.5 ± 0.50 mm) against *B. subtilis*, while *Streptomyces* sp. strain CALT_2 showed the most inhibitory activity against *C. albicans*; in general, all potential strains showed a significant variation in inhibitory activity against tested pathogens. However, the potential strains showed more inhibitory activity against Gram-positive than Gram-negative bacteria. In addition, the MICs of their extracts revealed that they had good inhibitory activity against Gram-positive, Gram-negative, and fungal pathogens. The MIC values ranged between 35 and 86 µg/mL (Figure 2, Figure 3 and Figure 4), and the results obtained were higher than those obtained from species of Streptomyces against tested pathogens by previous studies [37,38]. Further studies are needed in this unexplored desert area in regard to the discovery of novel microorganisms, especially Actinobacteria, and to their biotechnological applications in several fields.

An acute toxicity test of the *C. procera* ethanolic extract revealed that animals that received doses of up to 5000 mg/kg of the extract did not die or display the appearance of any signs of toxicity, indicating that the LD50 was higher than 5 g/kg. Based on previous studies [39], it has been proven that substances with a half-lethal dose higher than 5 g/kg are nontoxic substances, and therefore, the toxicity of *C. procera* was not detected.

Mutations caused by carcinogens in somatic cells contribute to genetic instability, which is an essential feature of carcinogenesis. Antigenotoxic agents prevent the development of DNA adducts, activate DNA repair mechanisms, and have antioxidant functions [40]. The present results indicated that the mean percentage of chromosomal aberrations induced with 20 mg CP/kg b.wt. reached 16.8% (*p* > 0.01), compared with 2.6% for the control. Additionally, treatment with CP induced 8.77% DNA fragmentation in liver cells compared with 2.97% in control animals. CP is characterized by its inactive form; once it reaches the liver, it converts to active metabolites and generates reactive oxygen species [41,42], which induces genetic alterations and chromosomal breakages, rearrangements, aneuploidies, and other mutagenic effects [43,44].

*C. procera*, as a natural extract, was examined to minimize CP’s genotoxicity in the bone marrow and liver cells of mice. The results revealed that pretreatment with *C. procera* significantly decreased the percentage of chromosomal aberrations and DNA fragmentation induced by CP. This activity may be due to some bioactive secondary metabolites observed in the GC–MS analysis of *C. procera* leaf extracts, such as α-amyrin, lupeol acetate, and linolenic acid. α- and β-amyrins have been documented to have antitumor, anti-inflammatory [45], and antioxidant properties [46]. Lupeol compounds have been observed to exert antioxidant action [47,48]. Prasad et al. [49] stated that mice treated with lupeol had significantly reduced aberrant cells, micronuclei, and cytotoxicity induced by benzo(a)pyrene and increased mitotic indices. Using the comet assay, Blasi et al. [50] demonstrated that linoleic acid was an effective antigenotoxic compound against ethyl methanesulfonate in human hepatoma (HepG2) cells.

## 4. Materials and Methods

### 4.1. Reagents and Chemicals

All of the chemicals and reagents used were analytical grade. Ethyl alcohol and ethyl acetate were used for preparing the extract from plant and bacteria respectively. Mueller–Hinton (MH) broth and starch–casein agar medium were acquired from Himedia (Mumbai, India). Also acquired were Sabouraud’s agar from (Oxoid, Lenexa, KS, USA) and potato dextrose agar from (Difco, Göteborg, Sweden).

### 4.2. Sample Collection and Identification of Plant

The leaves of *C. procera* and the soil from its rhizosphere were collected in January 2020 from the college campus of the University of Jeddah, Khulais Governorate, Saudi Arabia. The selected plant was identified and authenticated by Prof. Amna Saddiq, Biology Department, Faculty of Science, and Jeddah University on the basis of taxonomic characters with a voucher number UJH#012020. The leaf parts were cut into small pieces and shade-dried for seven days. The dried, carved fragments were powdered using a mechanical grinder and stored in an airtight container. Collection and processing of soil from rhizosphere were performed according to method described by McPherson et al. [51].

### 4.3. Preparation of C. procera Leaf Extract

A total of two hundred grams of shade-dried *Calotropis procera* leaves was coarsely powdered, charged into aspirator bottles, and allowed to soak in hydroethanol (75%) (1:3 plant material to solvent) for 72 h at room temperature. The extracts were filtered, and the pooled ethanolic extract was evaporated under reduced pressure using a rotary evaporator (Bioevopak Co., Ltd., Jinan, China), followed by lyophilization using a laboratory lyophilizer until complete dryness [52]. The extract was preserved at −20 °C for use in this study. The extractable components obtained from *Calotropis procera* leaf accounted for 7.16%. The extract yield (g/100 g) was calculated using the following equation: yield (%) = (W1 × 100)/W2, where W1 is the weight of the extract residue obtained after solvent removal and W2 is the weight of raw material collected.

### 4.4. Isolation of Rhizosphere Inhabiting Actinobacteria

Soil samples were first pretreated; soil pretreatment was required for inhibiting or eliminating unwanted microorganisms. Moist heat treatment was employed for the selection of various actinobacteria groups. One gram of soil sample was serially diluted at 1:10 to 1:1000 in sterile saline solution. Soil extract medium [53] with glycerol as carbon source and starch–casein agar medium [54] were used for isolation of actinobacteria. The pH of the media used was set to 7.2. Cycloheximide and nystatin (0.050 mg/mL) were added to the medium as antifungal agents [55,56]. Nalidixic acid (10 mg/L) was also used to inhibit the bacteria capable of overcrowding without affecting the growth of actinobacteria [57,58] Plates were incubated at 28 °C, and the number of colonies was determined after 7–14 days. The purified colonies were maintained on to starch–casein slants and kept in 20% glycerol at −20 °C as stock cultures.

### 4.5. Primary Screening of Isolated Actinobacteria

A total of seventeen pure isolates were screened for antimicrobial activity by the agar disc method described by Thakur et al. [59]. The potential isolates were selected for secondary metabolite extraction with ethyl acetate according to the methods described by Chakraborty et al. [60] for secondary screening by the agar well-diffusion method described by Kadriye et al. [61].

### 4.6. Characterization of the Isolates

According to *Bergey’s Manual of Determinative Bacteriology* [62], the purified isolates were identified to the genus level after direct microscopic observation at 1000× magnification for the aerial and substrate mycelial growth on coverslips inserted in starch–casein agar medium [63]. In addition, the colors of aerial and substrate mycelia and the diffusible pigments produced were visually determined.

### 4.7. Molecular identification of Actinomycetes Isolates

Genomic DNA was extracted from four actinomycetal strains that showed the best antimicrobial activity according to the method described by Hong et al. [64]. The 16S rRNA gene was amplified with a set of bacteria-universal primers (Invitrogen, USA); the primers 27F (5-GAGTTTGATCCTGGCTCA-3) and 1498R (5-ACGGCTACCTTGTTACGACTT-3), which are complementary to the conserved regions at the 5- and 3- ends of the *E. coli* 16S rRNA gene [65]; 3 mM MgCl_2_; 3 mM dNTPs; 5 µL of Taq buffer; and 1 U Taq DNA polymerase (Invitrogen, Waltham, MA, USA). PCR amplification was performed on a cycler PCR machine (Bio-Rad Laboratories, Segrate, Italy), with the initial denaturation at 94 °C for 5 min, followed by 50 cycles of amplification (94 °C for 1 min, 54 °C for 1 min, and 72 °C for 2 min) and an extension step (72 °C for 5 min). Of each PCR product, 50 ng/μL was used to prepare the samples, which were delivered to MacroGen Company in Korea (http://www.dna.macrogen.com, accessed on 20 October 2021) following their specifications. The sequences were analyzed using BLAST (http://www.ncbi.nlm.nih.gov/BLAST, accessed on 20 October 2021) to preliminarily identify the strains. The cluster analysis was performed using the MEGAX (10.1.8) software package.

### 4.8. In Vitro Antimicrobial Activity

The antimicrobial activity of both plant extracts and selected actinobacterial isolates extracts against the Gram-negative bacteria *E. coli* (ATCC 25955) and *K. pneumonia* (ATCC 13883), the Gram-positive bacteria *B. subtilis* (NRRL B-543) and *S. aureus* (ATCC 25923), the unicellular fungus *C. albicans* (ATCC 10231), and the filamentous fungus *Aspergillus fumigatus* (ATCC 1022) was determined by agar well-diffusion assay according to the method described by Balouiri et al. [66] based on the measurement of the diameter of the inhibition zone in mm. Mueller–Hinton agar (Merck) was used for the growth of bacterial test strains at 37 °C for 24 h, Sabouraud’s agar (Oxoid) for growth of unicellular fungi (yeast) at 30 °C for 24 h, and potato dextrose agar (Difco) for growth of the fungal strain at 28 °C for 48 h. Both plant and actinobacterial extracts of 10 mg/mL concentration were prepared in dimethyl sulfoxide (DMSO). Wells containing the same volume of DMSO (1%) served as negative controls. At the same time, the standard antibiotic ketoconazole (25 μg/mL) for fungi and gentamycin (12 μg/mL) for actinobacteria were used as positive controls. All treatments were performed in triplicate.

### 4.9. MIC Test

MICs for both ethanolic extract of *C. procera* leaves and ethyl acetate extract of actinobacteria isolates against test microbes, previously mentioned, were determined by the serial dilution method from 20 to 120 µg/mL (as 6 successive concentrations) according to the method described by Zgoda and Porter [67]. MIC values were recorded as the lowest concentration of the extract that inhibited the growth of the test pathogens [68].

### 4.10. Determination of Bioactive Compounds by Gas Chromatography–Mass Spectrometry Analysis

Analysis of the *C. procera* leaves and the most potential actinobacterial strain (CALT_2) extracts was carried out at the National Research Center in Cairo, Egypt using a GC–MS spectrometer (THERMO Scientific TRACE 1310 Gas chromatograph, Waltham MA, USA) with an ISQ Single Quadrupole Mass Spectrometer. The GC–MS scheme had a DB5/MS column (J & W Scientific, 30 m × 0.25 mm i.d., 0.25 μm film thickness) (Agilent, Santa Clara, CA, United States). Helium was used as the carrier gas at a flow rate of 1.5 mL/min and a split ratio of 1:10. Temperature programming was applied (50 °C for 1 min, 150 °C for 1 min, 250 °C for 5 min, and 290 °C for 10 min). The injector and detector were maintained at 250 °C. Diluted samples (1:10 diethyl ether, *v*/*v*) of 5 μL were injected. The total running time was 65 min. Mass spectra were obtained by electron ionization (EI) at 70 eV using a spectral range of *m*/*z* 40–450. The identity of each compound was determined by comparing its retention index with the spectra documented in the Wiley 9 database13.

### 4.11. In Vivo Antigenotoxic Activity of C. procera

#### 4.11.1. Experimental Animals

Swiss albino male mice 8–10 weeks old with an average weight of 27.5 ± 2.5 g were purchased from the National Research Center Animal House (Dokki, Cairo, Egypt). The animals were fed throughout the experimental period with a special powdered diet (protein: 160.4 g/kg, fat: 36.3 kg, fiber: 41 g/kg, with 12.1 MJ of metabolized energy) containing no inorganic sorbents purchased from Meladco Feed Co. (Aubor City, Cairo, Egypt) and housed in filter-top polycarbonate cages in a room free from any source of chemical contamination, artificially illuminated (12 h dark/light cycle), and thermally controlled (25 ± 1 °C). All animals were received humane care in compliance with the Animal Care guidelines as approved by the Committee of the National Research Center, Dokki, Cairo, Egypt and the National Institutes of Health (ethical code: NRC-NIH-86-23-1985).

#### 4.11.2. Determination of LD50 of *C. procera* Ethanolic Extract in Male Mice

Lorke’s method for determining acute toxicity (LD50) was used in this investigation [69]. The investigation included 2 phases. Nine mice were randomly divided into three groups of three mice each and given 10, 100, and 1000 mg extract/kg body weight orally in the first phase. The protocol was repeated in the second phase of the trial with three mice, each randomly divided into three groups of one mouse and administered 1600, 2900, or 5000 mg extract/kg body weight. At all phases, animals were observed for signs of adverse effects and mortality [70].

#### 4.11.3. Experimental Design

The animals were divided randomly into 11 groups (10 mice/group) after one week of acclimatization. Positive control animals were treated intraperitoneally with CP (20 mg/kg). Three groups of animals were treated orally using a gavage tube (gauge = 20) with 50 mg/kg of *C. procera* for 1, 7, and 14 days; three groups of animals were treated orally with 100 mg/kg of *C. procera* for 1, 7, and 14 days; and three groups of animals were treated orally with a high dose of *C. procera* (200 mg/kg b.wt) for 1, 7 and 14 days. The selected doses of the *C. procera* extract administered to the mice were according to the prescription of previous studies that used *C. procera* for as therapeutic agent for other disorders [70,71]. At the end of the treatment period, half of each treatment group (5 animals) was dissected for liver samples, and the other half was injected i.p. with colchicines 2 h before sacrifice. Bone marrow from the femur of each animal was obtained for the chromosomal aberration assay.

#### 4.11.4. Chromosome Abnormalities in Somatic Cell

Chromosome preparations from bone marrow were performed according to Rastrick [72]. Regarding the chromosomal abnormality of metaphases, one hundred well-spread patterns were analyzed per mouse. Metaphases with gaps, chromosome or chromatid breakage, and fragments were recorded.

#### 4.11.5. DNA Fragmentation Assay

DNA content was calorimetrically detected as described by Sahota et al. [73]. Hepatic tissue was dissociated in hypotonic lysis buffer (10 mM Tris, 1 mM EDTA, 0.2% Triton X-100, pH 8.0) and incubated for 30 min at 48 °C, and the intact chromatin (pellet) was separated from DNA fragments (supernatant) by centrifugation for 15 min at 12,000× *g*. The pellet was resuspended in a lysis buffer. Samples were precipitated with 10% trichloroacetic acid at 48 °C, pelleted at 4000 rpm for 10 min, mixed with 5% trichloroacetic acid, and boiled for 15 min, and DNA content was quantified using diphenylamine reagent. The percentage of DNA fragmentation was expressed using the following formula [74,75]:% DNA fragmentation = (O.D. of Supernatant/(O.D. of supernatant + O.D. of pellet)) × 100

#### 4.11.6. Statistical Analyses

Statistical analysis for the present data was performed using the SPSS V.20.0 software (SPSS Institute Inc., Cary, NC, USA). Student’s *t*-test was used to compare unpaired MICs of reference antibiotic and tested extract. The one-way analysis of variance (ANOVA) test was used to compare different groups in regard to the antigenotoxic activity of *Calotropis procera* leaf extract. Data were considered statistically significant at *p* < 0.05 [76].

## 5. Conclusions

The results of the current study indicated that both *C. procera* leaves and its rhizosphere-inhabiting actinobacterial strains are potential sources for antimicrobial metabolites. Considering the above results, one of the investigated strains, CALT_3, exhibited significant activity against pathogenic bacteria, while another, CALT_2, showed potential antifungal activity. Regarding the MIC assay, the actinobacteria inhabiting the rhizosphere of *C. procera* have the potential to be included in research of new preparations with antibacterial and antifungal action. Indeed, so does the plant extract, as it contains highly effective antimicrobial metabolites. Moreover, *C. procera* effectively reduced the genotoxicity induced by cyclophosphamide in a dose-dependent manner. However, further studies are required to determine the mechanisms involved in these antigenotoxic effects, which may contribute to a promising chemopreventive agent against carcinogenicity.

## 6. Research Limitations/Implications

The record of no death and signs of toxicity implied that the extract was safe for consumption even at a high dosage of 5000 mg/kg body weight. The significant reduction in chromosomal aberrations of the treated rats as compared with the control was an indication of antigenotoxic effect of the extract. The significant reduction evident of the extract at different days implies that the extract rate of lowering potentials was time dependent. The experiment had several limitations including small sample size and being performed on animal models in a relatively short time. The antimutagenic activity of tested extract remains unclear and will require further investigation.

## Figures and Tables

**Figure 1 molecules-27-03123-f001:**
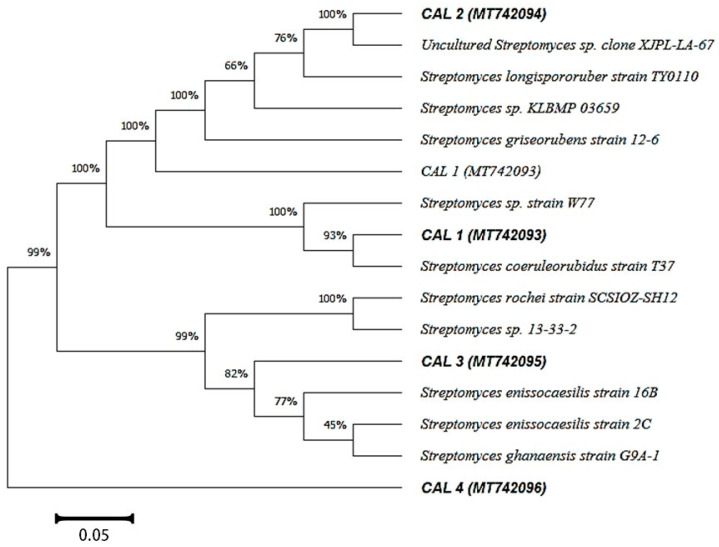
Neighbor-joining tree (partial sequences ~950 bp) showing the phylogenetic relationships of actinobacterial 16S rRNA gene sequences of potential strains to closely related (S ≥ 97%) sequences from the GenBank database.

**Figure 2 molecules-27-03123-f002:**
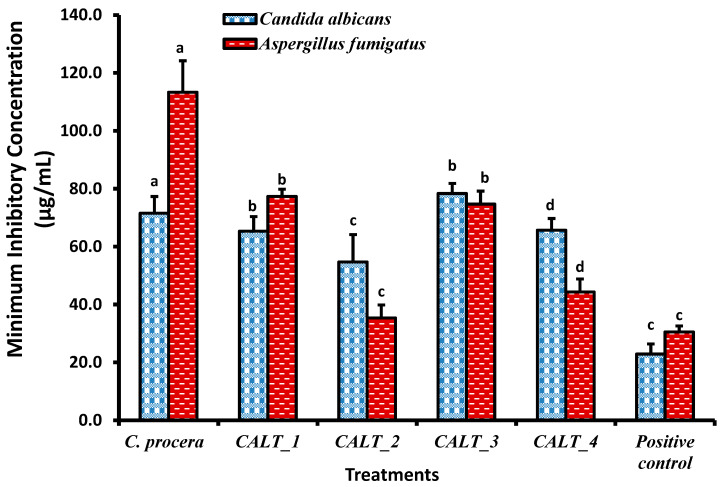
Minimum inhibitory concentrations (µg/mL) of *C. procera* ethanolic extract and actinobacteria isolate extracts against yeast and fungi. Values are mean ± SD; different letters (a, b, c, d) indicate significant differences.

**Figure 3 molecules-27-03123-f003:**
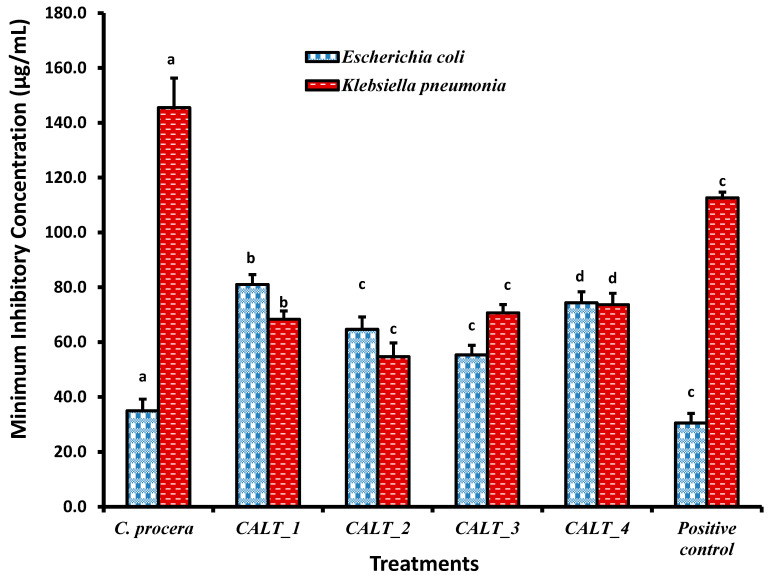
Minimum inhibitory concentrations (µg/mL) of *C. procera* ethanolic extract and actinobacterial isolate extracts against Gram-positive bacteria. Values are mean ± SD; different letters (a, b, c, d) indicate significant differences (*p* ≤ 0.05) between both plant and actinobacteria isolate extracts for the same pathogen according to one-way ANOVA test.

**Figure 4 molecules-27-03123-f004:**
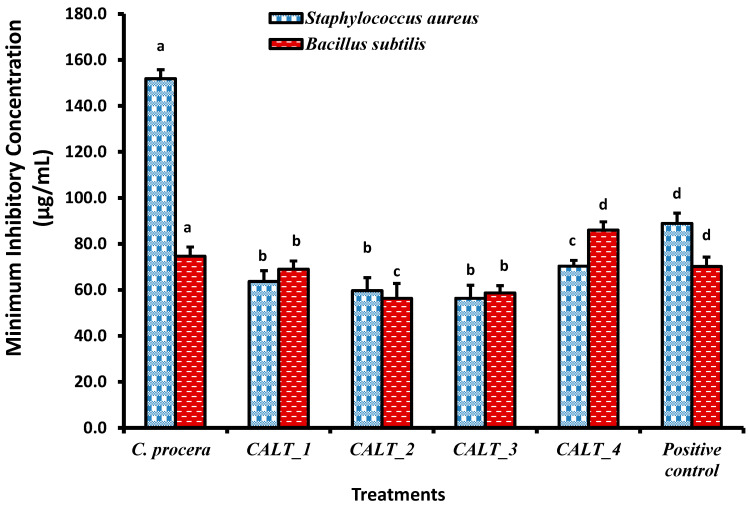
Minimum inhibitory concentrations (µg/mL) of *C. procera* ethanolic extract and actinobacterial isolates extract against Gram-negative bacteria. Values are mean ± SD; different letters (a, b, c, d) indicate significant differences (*p* ≤ 0.05) between both plant and actinobacteria isolate extracts for the same pathogen according to one-way ANOVA test.

**Figure 5 molecules-27-03123-f005:**
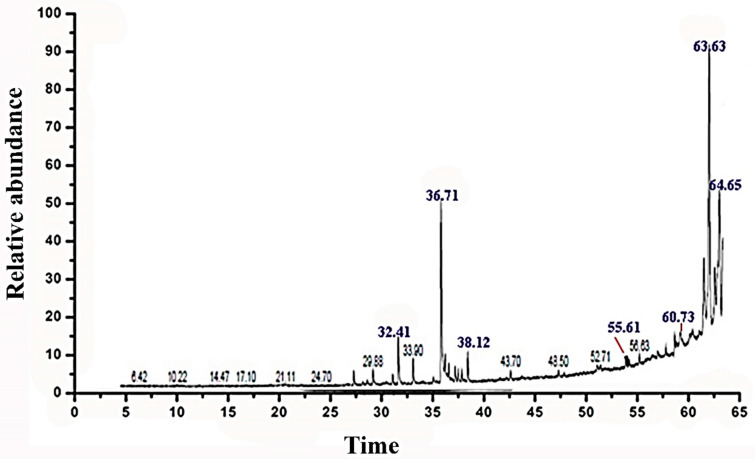
Chromatogram of ethanolic extract of *C. procera* leaves by gas chromatography–mass spectrometry (GC–MS) analysis. The GC–MS spectrum at retention time 0-65 min represents the main compounds.

**Figure 6 molecules-27-03123-f006:**
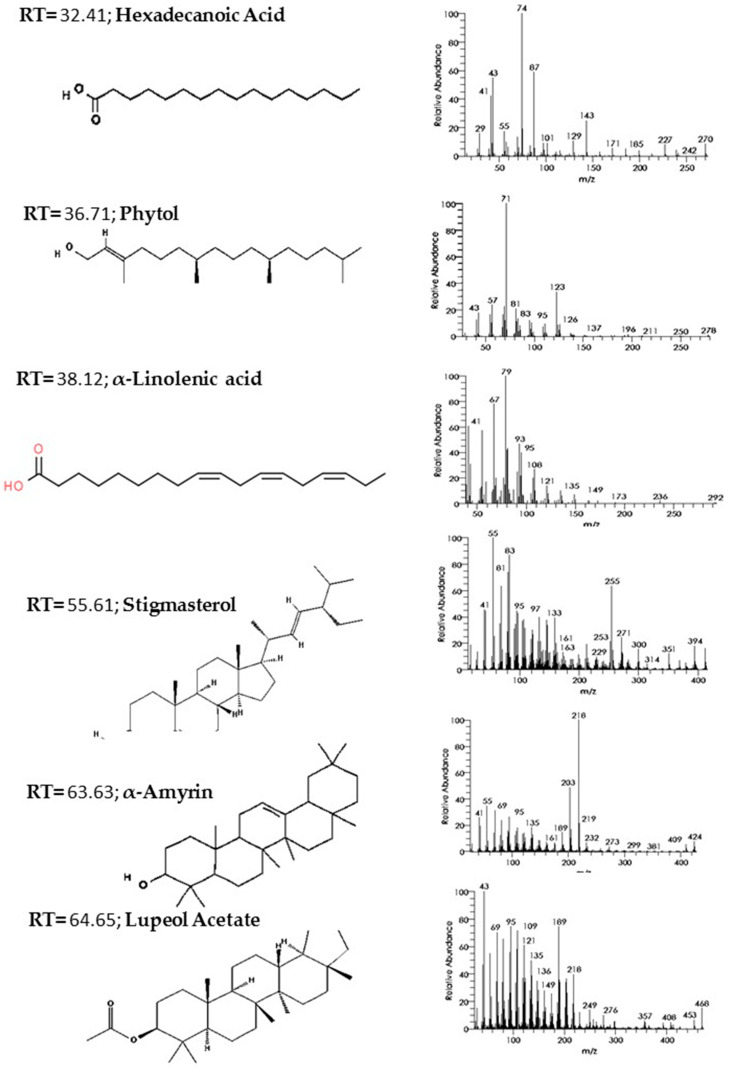
Hit spectra and chemical structures of the major phytochemical constituents of *C. procera* leaf extract. Chemical structure taken from ChemSpider web site, (accessed on 20 October 2021): hexadecanoic acid, phytol, stigmasterol, linolenic acid, α-amyrin, lupeol acetate.

**Table 1 molecules-27-03123-t001:** Zone of inhibition (mm) of ethanolic extract of *C. procera* leaves and rhizosphere-inhabiting actinobacterial isolates against pathogens tested using agar well-diffusion assay.

Antimicrobial Activity against Pathogens (Inhibition Zone in mm)						
Treatments (100 µg/mL)	Fungi (Yeast)		Gram-Positive		Gram-Negative	
	*C. albicans*	*A. fumigatus*	*S. aureus*	*B. subtilis*	*E. coli*	*K. pneumonia*
*C. procera*	21.00 ± 2.64 ^a^	12.00 ± 0.52 ^a^	18.66 ± 1.17 ^a^	16.23 ± 3.80 ^a^	21.93 ± 1.71 ^a^	21.26 ± 3.16 ^a^
CALT_1	10.43 ± 0.93 ^b^	7.17 ± 0.95 ^b^	8.16 ± 0.76 ^c^	7.43 ± 0.75 ^b^	7.83 ± 0.76 ^b^	9.94 ^b^ ± 0.81 ^b^
CALT_2	13.77 ± 1.36 ^b^	10.10 ± 1.85 ^b^	12.30 ± 1.30 ^b^	11.22 ± 0.96 ^c^	8.76 ± 0.92 ^b^	10.80 ± 0.43 ^b^
CALT_3	7.20 ± 0.82 ^c^	8.10 ± 2.25 ^b^	12.10 ± 2.01 ^c^	11.37 ± 1.30 ^c^	9.23 ± 0.25 ^b^	9.00 ± 0.60 ^b^
CALT_4	10.90 ± 0.90 ^b^	11.4 ± 1.00 ^a^	7.00 ± 0.10 ^b^	6.50 ± 0.50 ^b^	8.63 ± 0.66 ^b^	8.47 ± 0.62 ^b^
Positive control	20.33 ± 1.52 ^a^	11.16 ± 0.76 ^a^	16.53 ± 1.50 ^a^	20.07 ± 4.20 ^a^	23.40 ± 2.42 ^a^	20.20 ± 1.72 ^a^
Negative control	NI	NI	NI	NI	NI	NI

Values are mean ± SD; NI denotes no inhibition. Positive control for fungi: 25 µg/mL ketoconazole; positive control for bacteria: 25 µg/mL. Negative control: 50% of ethanol. For each column, the same letter shows that the difference between the means was not statistically significant. However, different letters show statistically significant differences (*p* < 0.05) between the corresponding treatments.

**Table 2 molecules-27-03123-t002:** Identification of phytocomponents of ethanolic extract of *C. procera* leaves using GC–MS analysis.

RT ^a^ (min)	Area %	Compound Name	Molecular Formula	Molecular Weight
27.95	0.91	Neophytadiene	C_20_H_38_	278
29.3	0.38	8-Heptadecyne, 1-Bromo-	C_17_H_31_Br	314
29.88	1.26	2-Pentadecanone	C_18_H_36_O	268
31.86	0.87	Tert-Hexadecanethiol	C_16_H_34_S	258
32.41	5.55	Hexadecanoic Acid	C_17_H_34_O2	270
35.93	0.55	7,9-Di-Tert-Butyl-1-oxaspiro (4,5) DeCa-6,9-Diene-2,8-Dione	C_17_H_24_O_3_	276
36.71	13.32	Phytol	C_20_H_40_O	296
37.12	1.23	17-Octadecenoic Acid,	C_19_H_36_O_2_	296
		Methyl Ester		
37.48	0.95	9,12-Octadecadienoic Acid (Z,Z)-,Methyl Ester	C_19_H_34_O_2_	294
38.12	3.04	α-Linolenic acid	C_19_H_32_O_2_	292
38.43	0.89	9-Octadecenoic Acid (Z)-,	C_20_H_38_O_2_	310
		Ethyl Ester		
38.78	0.89	Linoleic Acid Ethyl Ester	C_20_H_36_O_2_	308
43.69	0.75	À-D-Glucopyranoside, Methyl	C_16_H_32_BNO_6_Si	373
		2-(Acetylamino)-2-Deoxy-3-O-(TrimEthylsilyl)-, cyclic butylboronate		
48.51	0.5	Promecarb	C_16_H_16_N_2_O_5_	316
49.06	0.31	14-Hydroxy-14-Methyl-Hex	C_18_H_34_O_3_	298
		Adec-15-Enoic Acid Methyl		
		Ester		
52.4	0.54	Methanesulfonic Acid	C_26_H_43_DO_4_S	453
52.72	0.53	À-Tocospiro A	C_29_H_5_0O_4_	462
55.26	0.63	Picrotin	C_15_H_18_O_7_	310
55.61	3.16	Stigmasterol	C_29_H_48_O	412
56.63	1.61	Boroxin,	C_21_H_12_B_3_F_9_O_3_	516
57.9	0.68	Tetrakis (4-Methylphenyl) Thieno3,2-BThiophene	C_34_H_28_S_2_	500
58.47	0.52	Astilbin	C_21_H_22_O_11_	450
59.28	0.67	Thieno3,4-CPyridine,	C_31_H_21_NS	439
		1,3,4,7-Tetraphenyl-		
60.17	1.42	Nicotiflorin	C_27_H_30_O_15_	594
60.34	0.9	Momordicinin	C_30_H_46_O_2_	438
60.73	2.14	Gombasterol A	C_28_H_48_O_7_	496
61.96	0.66	25-Hydroxy-24-Epi-Brassinolide	C_28_H_48_O_7_	496
62.67	0.61	1,2-Dilinoleoyl-Sn-Glycero-3-Phosph Oethanolamine	C_41_H_74_NO_8_P	739
63.63	39.36	α-Amyrin	C_30_H_50_O	426
64.65	17.94	Lupeol Acetate	C_32_H_52_O_2_	468
64.97	0.27	Methyl Commate D	C_31_H_50_O_4_	486

**^a^** RT: retention time of the compounds based on GC–MS peaks; compounds are listed in order of their elution from a DB5/MS column.

**Table 3 molecules-27-03123-t003:** Identification of metabolites components produced by *Streptomyces* sp. strain CALT_2 using gas chromatography–mass spectrometry (GC–MS) analysis.

RT ^a^ (min)	Area %	Compound Name	Molecular Formula	Molecular Weight
32.41	44.48	Hexadecanoic acid	C_17_H_34_O_2_	270
55.61	18.48	Stigmasterol	C_29_H_48_O	412
63.63	39.8	α-Amyrin	C_30_H_50_O	426

**^a^** RT: retention time of the compounds based on GC–MS peaks; compounds are listed in order of their elution from a DB5/MS column.

**Table 4 molecules-27-03123-t004:** Number and mean percentages of different chromosomal aberrations in bone marrow cells of mice after treatment with different doses of *C. procera* for 1, 7, and 14 days.

Groups	*C. procera* Treatment Day(s)	No. of Metaphases with				Total Chromosomal Aberrations	
		Gap	Frag. and/or Break	Del.	Gap + (Frag. and/or Break)	Excluding Gaps Mean ± S.E.	Including Gaps Mean ± S.E.
Control	1	7	12	—	4	3.2 ± 0.23	4.6 ± 0.3
*C. procera* 50 mg/kg		9	10	1	4	3.0 ± 0.33	4.8 ± 0.3
*C. procera* 100 mg/kg		7	9	—	5	2.8 ± 0.21	4.2 ± 0.32
*C. procera* 200 mg/kg		9	11	1	3	3.0 ± 0.2	4.8 ± 0.5
Control	7	8	9	—	4	2.6 ± 0.24	4.2 ± 0.22
*C. procera* 50 mg/kg		8	10	—	4	2.8 ± 0.3	4.4 ± 0.3
*C. procera* 100 mg/kg		11	9	—	3	2.4 ± 0.22	4.6 ± 0.32
*C. procera* 200 mg/kg		9	11	1	2	2.8 ± 0.2	4.6 ± 0.22
Control	14	11	10	—	3	2.6 ± 0.2	4.8 ± 0.34
*C. procera* 50 mg/kg		12	11	—	1	2.4 ± 0.23	4.8 ± 0.24
*C. procera* 100 mg/kg		9	10	—	4	2.8 ± 0.23	4.6 ± 0.2
*C. procera* 200 mg/kg		7	9	—	3	2.4 ± 0.2	3.8 ± 0.2

The total number of scored metaphases was 500 (5 animals/group). Frag. = fragment; Del = deletion.

**Table 5 molecules-27-03123-t005:** Number and mean percentage of the different types of chromosomal aberrations in bone marrow cells of mice after treatment with *C. procera* 200 mg/kg for 1, 7, and 14 days alone or in combination with cyclophosphamide 20 mg/kg.

Groups	*C. procera* Treatment day(s)	No. of Metaphases with				Total Chromosomal Aberrations					Inhibition %
		Gap	Frag. and/or Break	Del	Rt	Gap + (Frag. and/or Break)	End	Poly	Excluding Gaps Mean ± S.E.	Including Gaps Mean ± S.E.	
Control (nontreated)	1	9	10	1	—	2	—	—	2.6 ± 0.4	4.4 ± 0.22	8.3 ^a^
CP		20	42	7	4	20	3	8	16.8 ± 0.33 ^a^	20.8 ± 0.36 ^a^	
*C. procera* + CP		21	40	6	3	21	2	5	15.4 ± 0.44 ^a^	19.6 ± 0.4 ^a^	
Control	7	10	9	1	—	1	—	—	2.2 ± 0.22	4.2 ± 0.21	42.85 ^b^
CP		20	42	7	4	20	3	8	16.8 ± 0.33 ^a^	20.8 ± 0.36 ^a^	
*C. procera* + CP		15	28	2	—	15	3	—	9.6 ± 0.86 ^b^	12.6 ± 0.8 ^b^	
Control	14	9	11	—	—	3	—	—	2.8 ± 0.3	4.6 ± 0.3	58.3 ^b^
CP		20	42	7	4	20	3	8	16.8 ± 0.33 ^a^	20.8 ± 0.36 ^a^	
*C. procera* + CP		16	24	2	—	9	—	—	7.0 ± 0.35 ^b^	10.2 ± 0.43 ^b^	

The total number of scored metaphases is 500 (5 animals/group). Frag. = fragment; Del = deletion; Rt. = Robertsonian translocation; End. = endomitosis; Poly. = polyploidy. ^a^ Significant at the 0.05 level (one-way ANOVA test) compared with control (nontreated). ^b^ Significant at the 0.05 level (one-way ANOVA test) compared with treatment.

**Table 6 molecules-27-03123-t006:** DNA fragmentation in mouse liver cells after treatment with different doses of *C. procera* for 1, 7, and 14 days.

Groups	Days	DNA Fragmentation
Control	1	2.92 ± 0.2
*C. procera* (50 mg/kg)		3.7 ± 0.46
*C. procera* (100 mg/kg)		3.15 ± 0.23
*C. procera* (200 mg/kg)		3.22 ± 0.03
Control	7	3.33 ± 0.29
*C. procera* (50 mg/kg)		2.98 ± 0.3
*C. procera* (100 mg/kg)		3.32 ± 0.34
*C. procera* (200 mg/kg)		3.17 ± 0.29
Control	14	3.2 ± 0.25
*C. procera* (50 mg/kg)		3.1 ± 0.23
*C. procera* (100 mg/kg)		3.32 ± 0.33
*C. procera* (200 mg/kg)		3.41 ± 0.2

**Table 7 molecules-27-03123-t007:** DNA fragmentation in mouse liver cells after treatment with *C. procera* 200 mg/kg for 1, 7 and 14 days alone or in combination with cyclophosphamide 20 mg/kg.

Groups	Days	DNA Fragmentation	DNA Fragmentation Inhibition %
Control	1	2.97 ± 0.27	
CP		8.77 ± 0.37 ^a^	
*C. procera* + CP		8.58 ± 0.38 ^b^	2.1 ^b^
Control	7	3.07 ± 0.22	
*C. procera* + CP		5.34 ± 0.3 ^b^	39.11 ^b^
Control	14	2.29 ± 0.2	
*C. procera* + CP		4.29 ± 0.23 ^b^	51.08 ^b^

^a^ Significant at the 0.05 level (one-way ANOVA test) compared with control (nontreated). ^b^ Significant at the 0.05 level (one-way ANOVA test) compared with CP.

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
