# Peer review of "Antimicrobial, Antigenotoxicity, and Characterization of Calotropis procera and Its Rhizosphere-Inhabiting Actinobacteria: In Vitro and In Vivo Studies"

_molecules, 2022, doi:10.3390/molecules27103123_

Round 1
Reviewer 1 Report
It is very-well marked that this study is acceptable wit minor revision and useful for publish in this journal.

Author Response
Reviewer 1
Comments and Suggestions for Authors
It is very-well marked that this study is acceptable wit minor revision and useful for publish in this journal.
Response: thank you. All your comments and suggestions mentioned in the attached PDF file were considered well.

Reviewer 2 Report
This work is a straightforward analysis of the antibacterial and antifungal properties of Calotropis procera leaves and of actinobacteria from its rhizosphere. The authors show that both the leaves and the bacteria extracts present these activities, with the bacterial isolates being more efficient. The data are clearly presented and convincing.
The authors find three compounds in extracts from one bacterial isolate that are also present in the plant's leaves, but do not really exploit this finding. What are the relative concentrations in leaves versus bacteria? It would be nice in the discussion to explore the possibility that the bacterial isolates may be responsible for the activity, or even the presence of active compounds in the plant's leaves. Are these potential endophytes, that could also be present in the leaves and thus confer resistance to the plant against infections?
Minor points:
line 19: "A total of 31 compounds were identified..." from what? Plant leaf extracts, I guess?
line 182: the alpha in a-amyrin is wrongly written on several occasions (also in Figure 6).
lines 362-364: Was "Moist heat treatment" the treatment applied? I actually don't see how to make sure to eliminate "unwanted microorganisms". This, combined to the addition of cycloheximide, nystatin and nalidixic acid seems quite a restrictive way of analyzing the rhizosphere microbiota.
line 336 and 338: "Lepoel": Is it Lupeol?
Reviewer 3 Report
The authors investigated ‘‘Antimicrobial, antigenotoxicity, and characterization of Calotropis procera and their rhizosphere-inhabiting actinobacteria: in vitro and in vivo studies’’. The proposed manuscript deals with an actual and interesting topic since drug resistance to infectious agents represents a serious problem and the use of medicinal plants represents an inexhaustible source for overcoming it. The appropriate methodology has been employed, and the obtained findings are interesting. However, the authors should make some corrections to improve the quality of this paper.
- The authors do not state information regarding the extraction yield analyzed extract.
- The authors should include a short paragraph regarding the chemical substances used in the study.
- In the Material and methods section the authors should involve information regarding the method of preparation of the extract for oral administration. How the extracts were applied, by gavage or dissolved in tap water?
- The preferable laboratory animal for acute oral toxicity test is a female rat, thus a rationale for using male rats should be provided.
- Please provide the rationale for dosage selection of Calotropis procera
- What was the reason for sacrifice animals on 14th day?
- Limitations of the study need to be added.
- Quality of images are poor and need to be improve.
